# Realization of a two-dimensional Weyl semimetal and topological Fermi strings

Qiangsheng Lu[1,2,13], P. V. Sreenivasa Reddy [3,13], Hoyeon Jeon [4,13], Alessandro R. Mazza[2,5], Matthew Brahlek [2], Weikang Wu [6], Shengyuan A. Yang[6], Jacob Cook[1], Clayton Conner[1], Xiaoqian Zhang[1], Amarnath Chakraborty[1], Yueh-Ting Yao[3], Hung-Ju Tien [3], Chun-Han Tseng[3], Po-Yuan Yang[3], Shang-Wei Lien[3], Hsin Lin [7], Tai-Chang Chiang [8,9], Giovanni Vignale [1], An-Ping Li [4] ✉, Tay-Rong Chang [3,10,11] ✉, Rob G. Moore [2] ✉ & Guang Bian [1,12] ✉

A two-dimensional (2D) Weyl semimetal, akin to a spinful variant of graphene, represents a topological matter characterized by Weyl fermion-like quasi-particles in low dimensions. The spinful linear band structure in two dimensions gives rise to distinctive topological properties, accompanied by the emergence of Fermi string edge states. We report the experimental realization of a 2D Weyl semimetal, bismuthene monolayer grown on SnS(Se) substrates. Using spin and angle-resolved photoemission and scanning tunneling spectroscopies, we directly observe spin-polarized Weyl cones, Weyl nodes, and Fermi strings, providing consistent evidence of their inherent topological characteristics. Our work opens the door for the experimental study of Weyl fermions in low-dimensional materials.

The discovery of Weyl semimetals, which host spin-split massless quasiparticles in three-dimensional (3D) crystals, is particularly thrilling as it represents an experimental realization of Weyl fermions, a concept proposed long ago in the realm of particle physics[1–5]. The chiral nodal points and 2D Fermi arc surface states of 3D Weyl semimetals bring about exotic properties such as chiral anomaly, unusual optical conductivity and nonlocal transport[6–16]. While the Weyl equation was derived only for odd spatial dimensions, the generalization of a 3D Weyl fermion state in 2D leads to a distinct topological state of matter, labeled as 2D Weyl semimetals, that exhibit a spin-polarized analog of graphene. The dimension

reduction gives rise to a multitude of unconventional physical properties, including parity anomaly in (2 + 1)-D (space-time) quantum field theory[17–20], charge fractionalization with zero modes of charge $e/2$[21], spin-valley Hall effects[22,23], giant Berry curvature dipole (BCD)[24,25], and topological quantum criticality[22]. Among the various topological Dirac/Weyl solid-state materials (see Fig. 1a), 2D Weyl semimetals stand out as the final frontier that has yet to be extensively explored in experiments. A winding number of $\pi$ can be obtained by integrating the Berry phase along a loop encircling each Weyl node[26]. The nonzero winding number can be regarded as the topological charge of 2D Weyl semimetals, which guarantees the

[1]Department of Physics and Astronomy, University of Missouri, Columbia, MO 65211, USA. [2]Materials Science and Technology Division, Oak Ridge National Laboratory, Oak Ridge, TN 37831, USA. [3]Department of Physics, National Cheng Kung University, Tainan 701, Taiwan. [4]Center for Nanophase Materials Sciences, Oak Ridge National Laboratory, Oak Ridge, TN 37831, USA. [5]Materials Science and Technology Division, Los Alamos National Laboratory, Los Alamos, NM 87545, USA. [6]Research Laboratory for Quantum Materials, Singapore University of Technology and Design, Singapore 487372, Singapore. [7]Institute of Physics, Academia Sinica, Taipei 11529, Taiwan. [8]Department of Physics, University of Illinois at Urbana-Champaign, 1110 West Green Street, Urbana, IL 61801-3080, USA. [9]Frederick Seitz Materials Research Laboratory, University of Illinois at Urbana-Champaign, 104 South Goodwin Avenue, Urbana, IL 61801-2902, USA. [10]Center for Quantum Frontiers of Research and Technology (QFort), Tainan 70101, Taiwan. [11]Physics Division, National Center for Theoretical Sciences, Taipei 10617, Taiwan. [12]MU Materials Science & Engineering Institute, University of Missouri, Columbia, MO 65211, USA. [13]These authors contributed equally: Qiangsheng Lu, P. V. Sreenivasa Reddy, Hoyeon Jeon. ✉e-mail: apli@ornl.gov; u32trc00@phys.ncku.edu.tw; moorerg@ornl.gov; biang@missouri.edu

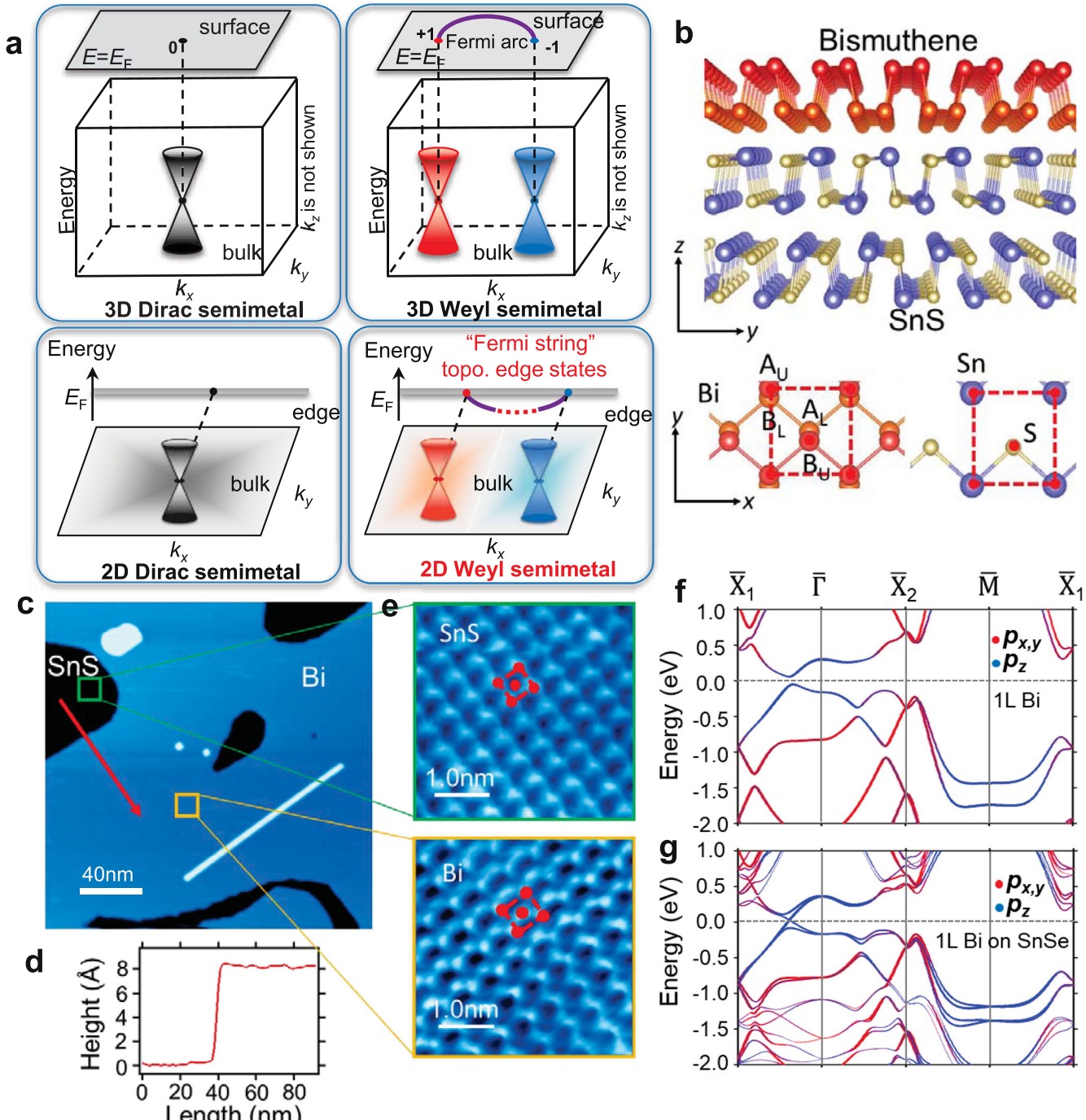

**Fig. 1 | Topology, lattice property, and band structure of 2D Weyl semimetal, α-bismuthene grown on SnS. a** Overview of Dirac/Weyl semimetals and their topological boundary states. **b** Side and top views of the lattice structure of bismuthene and SnS substrate. The red dashed squares indicate the unit cell of bismuthene and SnS. **c** Large-scale STM image of bismuthene grown on SnS substrate. **d** The height profile is taken along the red arrow in (**c**). **e** Zoom-in STM images of bismuthene (top) and the surface of SnS (bottom). The red dashed squares indicate the unit cell of SnS and bismuthene. **f** Calculated band structure of free-standing bismuthene. **g** Calculated band structure of bismuthene on SnSe.

existence of topologically protected edge states[27]. These topological edge states take the form of Fermi strings with one end attached to the projection of bulk Weyl nodes at the Fermi level, as schematically shown in Fig. 1a[28]. The Fermi string edge states serve as the 1D counterparts to the Fermi arc surface states observed in 3D Weyl semimetals. In this context, 2D Weyl semimetals present an unprecedented paradigm of bulk–boundary correspondence in topological materials. The highly unusual properties associated with 2D Weyl fermion states have inspired a myriad of theoretical and

experimental works[20,28–37]. Nevertheless, the realization of an intrinsic 2D Weyl semimetal in experiments remains elusive to date.

In this work, we report the realization of a 2D Weyl semimetal in an intrinsic 2D crystal, epitaxial bismuthene (a single atomic layer of bismuth stabilized in phosphorene structure[38,39]). Our spin- and angle-resolved photoemission spectroscopy (spin-ARPES) results unambiguously demonstrate the spin-polarized Weyl fermion states in bismuthene grown on SnS(Se) substrates. Furthermore, our scanning tunneling spectroscopy (STS) measurements unveil a robustly

enhanced local density of states at the edge of bismuthene, which aligns seamlessly with the calculated edge spectrum featuring Fermi string states. These experimental results establish epitaxial bismuthene on SnS(Se) as an ideal 2D Weyl semimetal.

## Results and discussion

### The material base for hosting 2D Weyl fermion states

Each Bismuth atom typically forms three covalent bonds with its neighbors. Two allotropic structural phases of bismuthene exist in the 2D limit, namely, the orthorhombic phosphorene-like phase[40,41] and the hexagonal honeycomb-like phase[42,43]. Here we grow the phosphorene-like bismuthene (bismuthene for short in the following) by molecular beam epitaxy (MBE). We selected SnS(Se) as the substrate due to its van der Waals semiconductor properties, and the (001) surface of SnS(Se) shares a lattice structure similar to that of bismuthene. Figure 1b schematically shows the lattice structure of the sample. Bi atoms form a single-layer phosphorene structure on the (001) surface of SnS(Se). The Bi atoms exhibit a pronounced $sp^3$-hybridization, resulting in the three Bi–Bi bonds adopting a nearly tetrahedral configuration. This configuration leads to two atomic sublayers, which are distinguished by red and orange colors in Fig. 1b. The unit cell has four atoms labeled as $A_U$, $A_L$, $B_U$, and $B_L$ (see the top view in Fig. 1b), where "U" and "L" indicate the upper and lower sublayer, respectively. The (001) surface of SnS(Se) shares the same lattice structure with bismuthene, with S(Se) occupying A sites and Sn occupying B sites. We performed scanning tunneling microscope (STM) measurements to map the surface topography of the Bi sample grown on SnS(Se). The STM result in Fig. 1c indicates the high structural quality of bismuthene. The apparent height of bismuthene (including the thickness of bimuthene and the spacing between bismuthene and SnS(001) surface) is 8.0 Å, which can be seen from the height profile in Fig. 1d. The zoom-in STM images in Fig. 1e demonstrate the surface unit cell of bismuthene and SnS(Se). We found the in-plane lattice constants $(a, b)$ are (4.5, 4.8 Å) for bismuthene, (4.1, 4.5 Å) for SnS surface, and (4.3, 4.6 Å) for SnSe surface, respectively.

### Band structure of the 2D Weyl semimetal

Here we present the first-principles band structures and angle-resolved photoemission spectroscopy (ARPES) spectra to verify the existence of the 2D Weyl fermion states in the epitaxial bismuthene films. The calculated band structure of free-standing bismuthene is shown in Fig. 1f. The only prominent band feature near the Fermi level is a gapped Dirac cone located at a generic $\boldsymbol{k}$ point between $\bar{\Gamma}$ and $\bar{X}_1$. The gapped Dirac bands originated from $p_z$ orbitals of Bi atoms. The effective $\boldsymbol{k} \cdot \boldsymbol{p}$ model around the Dirac point can be written as $\mathcal{H}_0^{\text{Dirac}}(\boldsymbol{k}) = \tau_\pm (v_x k_x \sigma_x + \Delta k_x) + v_y k_y \sigma_y + \tau_\pm \lambda_{\text{SOC}} \sigma_z s_z$, where $(k_x, k_y)$ are measured from the Dirac nodes at $(\pm k_0, 0)$, $\sigma_i$ $(i = x, y, z)$ are Pauli matrices with respect to the basis of $\{|A, p_z\rangle, |B, p_z\rangle\}$ (the $p_z$ orbitals at the two sublattices A and B), $s_i$ $(i = x, y, z)$ are the spin matrices, $\Delta$ describes tilting of the Weyl cone in the $k_x$ direction and is crucial for generating a non-zero Berry curvature dipole (BCD)[24], $\lambda_{\text{SOC}}$ is the effective spin–orbit coupling, $\tau_\pm = \pm 1$ represents the chirality of the Dirac nodes located at $(\pm k_0, 0)$, and $v_{x,y}$ are Fermi velocity along $k_x$ and $k_y$ directions, respectively. $v_x = 3.17 \times 10^5$ m/s, $\Delta = 0.19 \times 10^5$ m/s, $v_y = 4.23 \times 10^5$ m/s, $\lambda_{\text{SOC}} = 55$ meV according to the first-principles results. The energy gap induced by spin-orbit coupling is $E_{\text{gap}} = 2\lambda_{\text{SOC}} = 0.11$ eV. Every band is doubly degenerate to the spin degree of freedom since the lattice of bismuthene is centrosymmetric.

The calculated band structure of bismuthene on SnSe is plotted in Fig. 1g. The presence of SnSe substrate breaks space-inversion symmetry and causes spin splitting in the bands of bismuthene. Specifically, the electronegativity difference between Sn and Se induces an in-plane dipole electric field, leading to an average energy shift of A sites relative to B sites of bismuthene. In addition, the surface potential of SnSe generates an electric field perpendicular to the surface, creating a

potential difference between the two Bi sublayers. The substrate perturbations can be effectively expressed as $\lambda_{\text{Dip}} \sigma_z + \lambda_V \sigma_y s_x + \tau_\pm \lambda_V' \sigma_x s_y$, where $\lambda_{\text{Dip}}$ describes the in-plane dipole field on the SnSe surface and $\lambda_V(\lambda_V')$ is the Rashba coupling caused by the electric field perpendicular to the substrate surface (see Supplementary Information for details). The spin splitting eliminates the SOC-induced energy gap, leading to the formation of a linear band crossing around the Fermi level. The 2D Weyl states generated through this method represent a fine-tuned critical state at the transition between two topologically distinct gapped phases. A detailed discussion on the formation mechanism of 2D Weyl states can be found in Supplementary Information-Section II. The states originating from the SnSe substrate are exclusively positioned either 0.9 eV below or 0.4 eV above the Fermi level. This unique arrangement can be attributed to the semiconductor nature of SnSe, characterized by a band gap of 1.3 eV. As a consequence, the 2D Weyl fermion states near the Fermi level primarily arise from Bi orbitals, confining them spatially within the Bi monolayer. This characteristic defines the Weyl fermion states as inherently 2D in nature.

The calculated 2D Weyl band structure is confirmed by our ARPES measurements. The ARPES results taken from the bismuthene/SnSe(S) sample are plotted in Fig. 2. The Fermi surface (Fig. 2a) contains two circular electron pockets in the direction of $\bar{X}_1 - \bar{\Gamma} - \bar{X}_1$. (We note that a similar pair of electron pockets show up in the direction of $\bar{X}_2 - \bar{\Gamma} - \bar{X}_2$ but with much lower intensity. This extra pair of pockets is due to the existence of Bi domains rotated by 90° in the MBE sample.) The ARPES spectra of bismuthene on SnSe along the lines of "cut1" and "cut2" are plotted in Fig. 2b, e. In the ARPES spectra, we found the band dispersion along "cut1" ("cut2") as well as that from a rotated Bi domain. This can be better seen in Fig. 2c, f with overlays of calculated bands on top of the ARPES spectra. The magenta lines are bands along "cut1" ("cut2"), while the green lines are bands along a direction perpendicular to "cut1" ("cut2"). The sole presence of linear Weyl cones near the Fermi level indicates that the transport and optical properties of this system are exclusively determined by the low-energy Weyl fermion states. No apparent gap was found at the nodal point, as evidenced by the second derivative spectrum (Fig. 2h) and the map of energy distribution curves (Fig. 2i). We notice that one linear branch of the Weyl cone is much dimmer than the other in the spectrum along "cut1" due to the photoemission matrix element effects. Figure 2d and g show the calculated spectra with the inclusion of the photoemission matrix elements, which agrees well with the ARPES result. The bismuthene/SnSe sample is electron-doped as the nodal point lies 0.1 eV below the Fermi level. We also performed ARPES measurements on bismuthene grown on SnS, and the results are plotted in Fig. 2j–l. Compared with bismuthene/SnSe, the Fermi level of bimuthene/SnS lies right at the Weyl node. The shift of the Fermi level can be attributed to the electronegativity difference and the doping deference between SnSe and SnS substrates. So, bismuthene/SnS is a perfect 2D Weyl semimetal with charge neutrality. Considering the different surface conditions of SnSe and SnS, the observation of gapless Weyl cones in both sample configurations indicates the robustness of the 2D Weyl fermion states against weak perturbations.

### Spin texture of the 2D Weyl cone

The defining characteristic of Weyl fermion states is the spin polarization of the linear band dispersion. The dipole term ($\propto \lambda_{\text{Dip}}$) together with the SOC term cause the spin to be polarized in the $z$ direction while the vertical-field terms ($\propto \lambda_V$ and $\lambda_V'$) give rise to an in-plane spin polarization (see Supplementary Information). As a result, a canted spin texture is expected for the Weyl cone. The calculated spin-resolved band spectrum along "cut1" is plotted in Fig. 3b. To verify the spin texture of the Weyl fermion states, we performed spin-resolved ARPES measurements on bismuthene/SnSe. Spin-resolved momentum distribution curves (MDC) along "cut1" at $E = -0.1$ eV are shown in

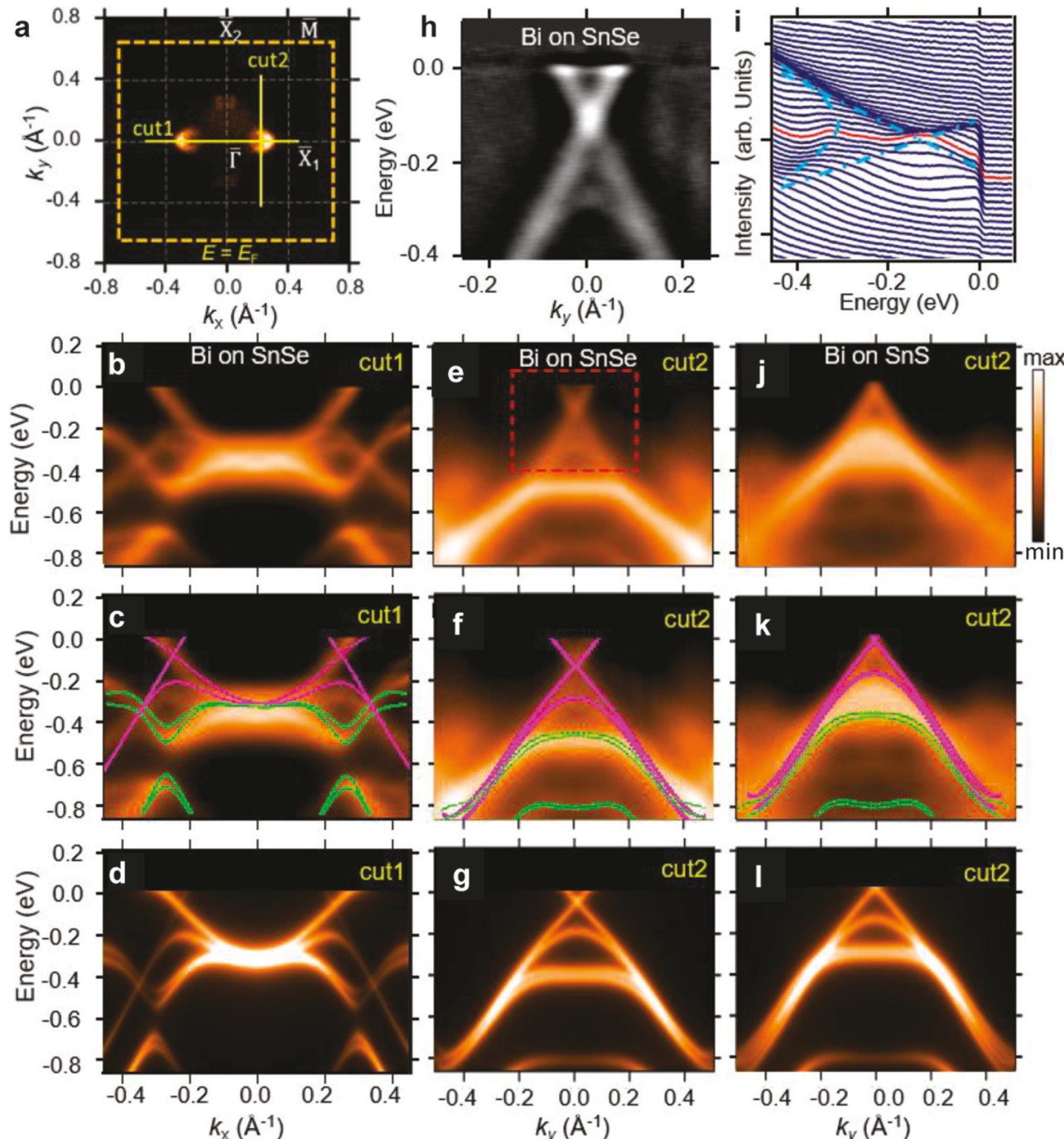

**Fig. 2 | ARPES and first-principles band structure of Weyl fermion states in epitaxial bismuthene. a** ARPES Fermi surface taken from bismuthene on SnSe. **b** ARPES spectra of bismuthene taken along the line of "cut1" marked in (**a**). **c** Overlay of calculated band structure on the ARPES spectrum along "cut1". The magenta lines are bands along the direction of $\overline{\Gamma}$-$\overline{X}_1$ while the green lines are bands in the direction perpendicular to $\overline{\Gamma}$-$\overline{X}_2$. **d** Calculated band spectra with the inclusion of photoemission matrix elements. **e**–**g** Same as **b**–**d** but for bands of bismuthene

on SnSe along the line of "cut2", which passes across the Weyl node. **h** Second derivative of the ARPES spectrum in the red box in (**e**). **i** Energy distribution curves (EDC) from the ARPES spectrum inside the red box in (**e**). The blue dotted lines mark the maximum of each EDC. The red solid line plots the EDC taken at the momentum of the Weyl point. **j**–**l** Same as **b**–**d**, but for bands of bismuthene on SnS along "cut2".

Fig. 3c. The blue and red lines are photoemission intensity recorded in the "spin-down" and "spin-up" channels for the corresponding spin component, respectively. The spin polarization can be calculated from the spin-ARPES spectrum according to the formula, $P = \frac{1}{S_{eff}} \frac{I_+ - I_-}{I_+ + I_-}$, where the effective Sherman function $S_{eff} = 0.275$ for our spin detectors[44–46]. The spin polarization extracted from the results in Fig. 3c is shown in Fig. 3d. The observed spin polarization of $\langle s_y \rangle$ is in good agreement with the theoretical result. The $s_x$ component is absent along the line of

"cut1", because the $\overline{\Gamma} - \overline{X}_1$ direction corresponds to a glide mirror of the lattice. We observed that the two valleys of Weyl fermion states exhibit opposite spin polarizations, a consequence of their partnership under time-reversal symmetry. The spin-integrated and spin-resolved ARPES spectra along "cut2" are plotted in Fig. 3e and f. The red and blue dots in Fig. 3f represent the "spin-up" and "spin-down" signals, respectively, recorded in the spin detector. This 2D "snapshot" of the $s_y$ component of the Weyl cone aligns with the calculated result

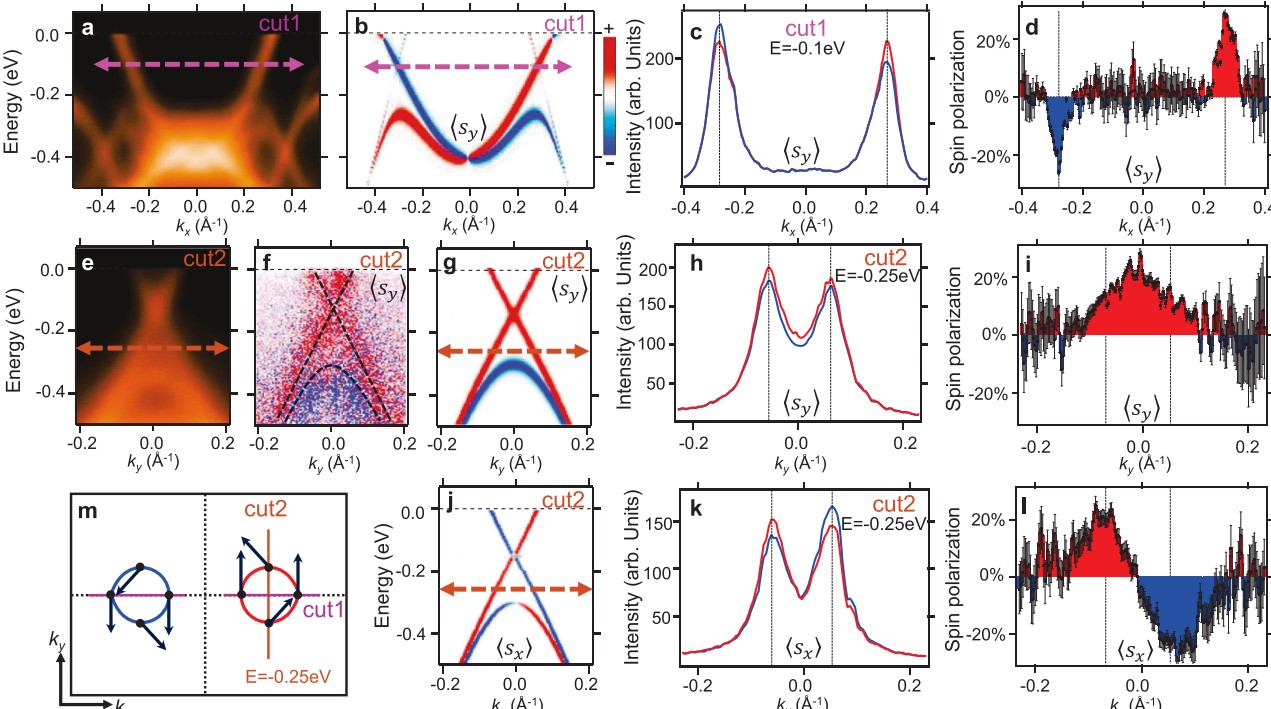

**Fig. 3 | Spin texture of 2D Weyl fermion states. a** ARPES spectrum along "cut1" marked in Fig. 2a. **b** Calculated band structure along "cut1" with the inclusion of photoemission matrix elements. The bands are colored according to the calculated expectation value $\langle s_y \rangle$ of each state. **c** Spin-resolved momentum distribution curves (MDC) taken at $E = -0.1$ eV along the line marked by the pink dashed arrows in (**a**) and (**b**). The blue and red curves are the photoelectron intensity recorded in the "spin-down" and "spin-up" channels of the spin detector, respectively. **d** The spin polarization extracted from the MDCs in (**c**). The shaded area indicates net spin polarization of $\langle s_y \rangle$. The formula for error bars can be found in the Supplementary Information. **e** ARPES spectrum along "cut2" marked in Fig. 2a. **f** 2D spin-resolved

ARPES map of $\langle s_y \rangle$ taken along "cut2". The blue and red dots represent the photoelectron signals recorded in the "spin-down" and "spin-up" channels of the spin detector, respectively. **g** Calculated electron band structure along "cut2", which is colored according to the expectation value $\langle s_y \rangle$ of each state. **h** Spin-resolved MDCs of $\langle s_y \rangle$ taken at $E = -0.25$ eV along the line marked by the green dashed arrows in (**e**) and (**g**). **i** Spin polarization of $\langle s_y \rangle$ extracted from the MDCs in (**h**). **j-l** Same as **g-i**, but for the spin component $\langle s_x \rangle$. **m** In-plane spin texture of iso-energy contours at $E = -0.25$ eV. The red and blue circles schematically depict the iso-energy contours of two valleys at $E = -0.25$ eV. The black arrows indicate the in-plane spin orientation of the Weyl fermion states.

presented in Fig. 3g, where the two linear branches of the Weyl cone display the same sign in the $s_y$ component. This characteristic is also evident in the spin-resolved MDC curves taken at $E = -0.25$ eV in Fig. 3h and the extracted spin polarization in Fig. 3i. In contrast, the two linear branches along "cut2" exhibit opposite signs in the $s_x$ component, as shown in the theoretical spin spectrum (see Fig. 3j). This spin behavior is confirmed by the experimental results of spin-resolved MDCs (Fig. 3k) and spin polarization (Fig. 3l). The in-plane spin texture of the Weyl fermion states is schematically summarized in Fig. 3m. We also measured the out-of-plane spin component. The magnitude of the observed spin polarization $\langle s_z \rangle$ is suppressed due to the existence of rotated domains in the MBE samples. Nonetheless, the observed $\langle s_z \rangle$ exhibits a similar characteristic to the $\langle s_y \rangle$ component, consistent with the first-principles results (see the Supplementary Information for details). Our spin-ARPES results demonstrate the unique spin texture of Weyl fermion states. This distinctive spin texture stands as a defining feature that sets apart 2D Weyl semimetals from Dirac semimetals like graphene.

## Bulk–boundary correspondence

To show the unique bulk–boundary correspondence of 2D Weyl semimetals, we calculated the bands of a semi-infinite bismuthene/ SnSe heterostructure with an open boundary in the (010) direction. The result is plotted in Fig. 4a. The Fermi string edge band emanates from the bulk Weyl nodes as required by the band topology of the 2D Weyl cone. More precisely, the Fermi string band comprises two chiral edge bands, each corresponding to the critical phase of a specific spin

sector. To illustrate this property, we utilized a tight-binding model incorporating $s_z$ conservation to simulate the edge electronic structure of the 2D Weyl semimetal. The edge spectrum is shown in Fig. 4b. In this case, the Weyl bulk bands comprise two independent spin sectors with $s_z = +\frac{1}{2}$ and $-\frac{1}{2}$, given that $s_z$ is a good quantum number. Each spin sector (colored in red and blue, respectively) corresponds to the critical phase of Haldane's quantum anomalous Hall model[47]. Consequently, each sector possesses a chiral edge band that disperses from the bulk nodal point at one valley to the bulk band edge at the other valley. Interestingly, the two edges demonstrate distinct connection patterns between the edge states and the bulk band. The Fermi strings connect to the valence band at the top edge and to the conduction band at the bottom edge, in accordance with the requirement of charge conservation. In real materials with strong spin–orbit coupling like bismuthene, $s_z$ is not a good quantum number, and thus the two spin sectors are allowed to hybridize with each other. On the other hand, the connection of Fermi string edge bands to the Weyl nodes remains topologically protected, because the winding number of linear bands is unaffected by spin–orbit coupling. (It is worth noting that, in addition to the Fermi string bands, there are additional edge state bands located within the bulk band gap (in Fig. 4a). The existence of those additional edge state bands can be attributed to the fact that the 2D Weyl semimetal is at a critical point in connection to two topologically distinct insulator phases (see Supplementary Information-Section XI for a detailed discussion).) The Fermi string edge bands give rise to an enhanced local density of states (LDOS) at the edge, especially, in a narrow energy window around the energy of bulk Weyl

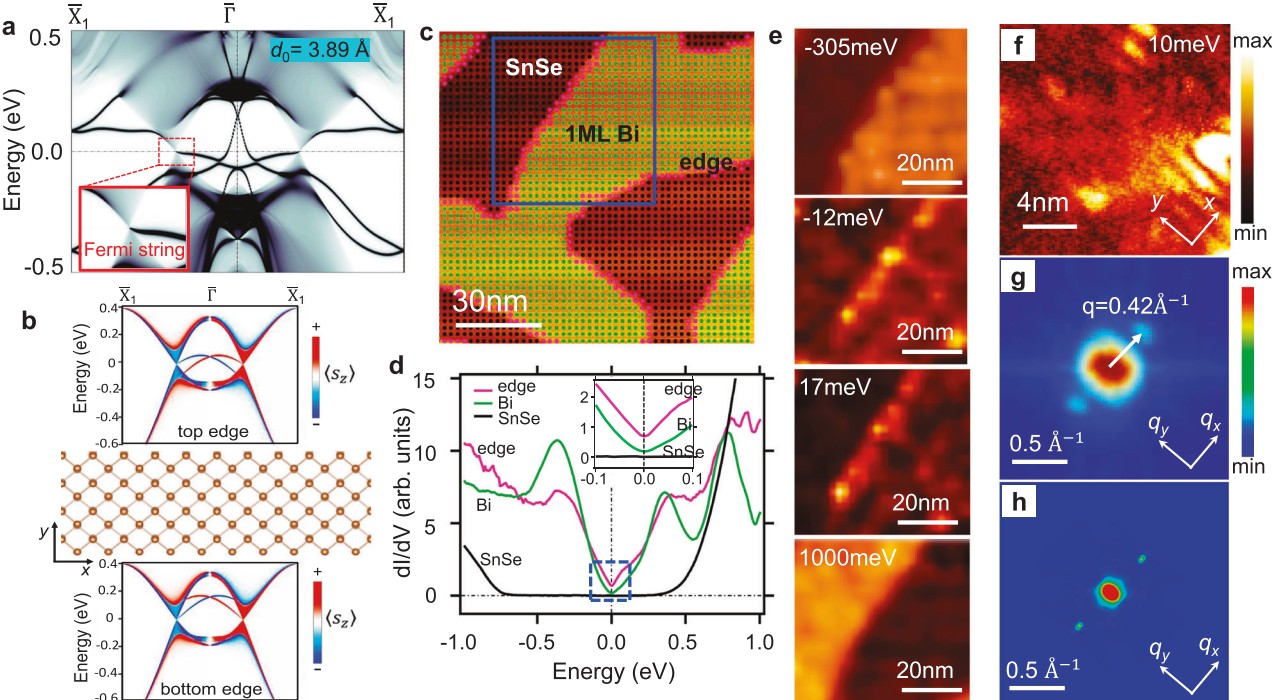

**Fig. 4 | Bulk–boundary correspondence in 2D Weyl semimetals. a** The edge bands and projected bulk bands of a semi-infinite bismuthene film on SnSe with an open boundary in the (010) direction. The bands are weighted with the charge density near the edge. $d_0 = 3.89\,Å$ is the relaxed interlayer spacing between bismuthene and SnSe substrate. **b** The connection of Fermi string edge bands to bulk Weyl nodes. The band structure was calculated by using a tight-binding model in which $s_z$ is conserved. **c** STM topography of bismuthene on SnSe. **d** Differential conductivity dI/dV spectra with the bias voltage aligned with the energy of Weyl nodes. The green curve is the averaged spectrum taken at the green grid inside the bismuthene patches shown in (**c**). The black curve is the averaged spectrum from the black grid on the surface of SnSe. The magenta curve is the averaged spectrum from the magenta points at the edges of the bismuthene patches. The inset shows the zoom-in dI/dV curves around the energy of Weyl nodes. **e** dI/dV map over the area marked by the blue box in **c** at bias voltage $V = -305, -12, 17$, and 1000 meV. **f** High-resolution dI/dV map at bias voltage $V = +10$ meV. **g** The corresponding Fourier transform of the dI/dV map in **f**, showing the quasiparticle interference (QPI) pattern. **h** The calculated quasiparticle interference pattern with spin-dependent scattering probability is considered.

nodes. The LDOS can be directly probed by the differential conductivity, dI/dV spectrum in STS experiments. An STM topography of bismuthene on SnSe is shown in Fig. 4c. The surface of SnSe, the interior of bismuthene, and the edge of bismuthene are marked by black, green, and magenta dots, respectively. We measured the averaged dI/dV spectrum at 4.6 K from the three regions (black, green, and magenta), and the result is plotted in Fig. 4d. A large gap of ~1.3 eV is observed in the dI/dV curve from the SnSe surface, indicating the Weyl fermion states are entirely confined within the Bi overlayer. Remarkably, the edge dI/dV spectrum (magenta) shows a notably enhanced LDOS compared to the bulk spectrum (green) in a narrow energy window around $E_W$ (the inset in Fig. 4d). This enhanced LDOS can also be seen in the dI/dV maps taken around a step edge of bismuthene (Fig. 4e). The edge appears brighter than both the SnSe surface and the interior of bismuthene only at $V_{bias} = -12$ and 17 meV, which correspond to energies close to $E_W$. A similar edge dI/dV spectrum was also observed in bismuthene/SnS samples at different temperatures (see Supplementary Information). The results consistently affirm the presence of Fermi string edge states near the energy of Weyl nodes.

The high-resolution dI/dV map unveils a unique plane wave-like pattern near the energy of Weyl nodes, as illustrated in Fig. 4f. The Fourier transform of the dI/dV map produces the quasiparticle interference pattern (Fig. 4g), featuring an oval contour at the center of the **q** space along with two smaller satellite contours. The separation between the satellite and the center of the oval, $\Delta q = 0.42\,Å^{-1}$, precisely matches the separation between the two Weyl nodes measured from ARPES results (Fig. 2a). Hence, the central oval contour in the QPI arises from intravalley scatterings, while the two satellites are induced by intervalley scatterings. The experimental QPI is consistent with the calculated result

shown in Fig. 4h. The plane wave-like QPI, characterized by a single wavevector, reflects the nodal Fermi surface of the 2D Weyl semimetal.

The 2D Weyl semimetal establishes a solid-state realization of Weyl fermions in 2D space. The two spin-polarized valleys in the epitaxial 2D Weyl semimetal correspond to two Weyl fermions with opposite helicities. Our demonstration of the 2D Weyl semimetal and the Fermi string edge states paves the way for the exploration of fascinating topological quantum properties of Weyl fermions in reduced dimensionality.

## Methods

### Growth of bismuthene on SnS and SnSe

Bi was deposited on the cleaved surface of SnS and SnSe crystals in an MBE-ARPES-STM ultrahigh vacuum (UHV) system. The SnS and SnSe crystals are n-type doped with Br. The base pressure was lower than $2 \times 10^{-10}$ mbar. High-purity Bi was evaporated from a standard Knudsen cell with a flux of 0.3 Å/min. The temperature of the substrate was kept at 50 °C during the growth. The substrate temperature is critical for growing a smooth Bi monolayer in the phosphorene structure.

### Scanning tunneling microscopy measurement

An in-situ Aarhus-150 STM was used to characterize the surface topography and the lattice parameters of the $\alpha$-Bi films. The topography was measured under room temperature, and the base pressure was lower than $2 \times 10^{-10}$ mbar. The bias voltage and the tunneling current were set to be 1.5 V and 0.01 nA for the surface topography measurement, 5 mV and 0.15 nA for the zoom-in atom-resolved STM measurement. The dI/dV spectrum, dI/dV mapping, and the STM QPI were produced by an Omicron LT-Nanoprobe system at 4.6 K. The sample

was transferred through an ultra-high vacuum (UHV) suitcase with pressure below $1 \times 10^{-9}$ mbar. The tunneling current is set to be 200 pA during the d$I$/d$V$ measurement.

## Spin- and angle-resolved photoemission spectroscopy measurements

Spin and angle-resolved photoemission spectroscopy measurements were performed in a lab-based system coupled to the molecular beam epitaxy system, using a Scienta DA30L hemispherical analyzer with a base pressure of $P < 5 \times 10^{-11}$ mbar and a base temperature of $T \sim 8$ K. Samples were illuminated with linearly polarized light using an Oxide $h\nu = 11$ eV laser system. The light polarization was set perpendicular to the sample and the slits of the detector. For electronic dispersion measurements, a pass energy of 2 eV and 0.3 mm slit was used for a total energy resolution ~2.5 meV and momentum resolution ~0.01 Å$^{-1}$. Dual VLEED ferrums that utilize exchange scattering are coupled to the electron analyzer and used to determine the spin $s_x$, $s_y$, and $s_z$ polarizations of the measured electrons. For spin-resolved measurements, a pass energy of 10 eV and a 1 mm $\times$ 2 mm spin aperture was used, yielding a total energy resolution ~50 meV and momentum resolution ~0.033 Å$^{-1}$.

## First-principles calculation

First-principles calculations with density function theory (DFT) were performed by using the Vienna ab Initio Simulation Package (VASP) package[48]. The Perdew–Burke–Ernzerhof (PBE)[49] exchange-correlation functional was used. The experimental lattice parameter was applied for bismuthene; the lattice parameter of SnSe(S) was modified to match the covered Bi. The spin–orbit coupling (SOC) was included self-consistently in the calculations of electronic structures with a Monkhorst–Pack $11 \times 11 \times 1$ $k$-point mesh. The vacuum thickness was >20 Å to ensure the separation of the slabs. Atomic relaxation was used until the residual forces were <0.01 eV/Å.

We constructed a tight-binding Hamiltonian for Bi/SnSe(S), where the tight-binding model matrix elements were calculated by projecting onto the Wannier orbitals[50–52], which used the VASP2WANNIER90 interface[53]. The Bi $p$ orbitals, Sn $p$ orbitals, and Se $p$ orbitals were used to construct the Wannier functions without performing the maximizing localization. The edge state electronic structure was calculated by Green's function technique, which computes the spectral weight near the edge of a semi-infinite system. To simulate the photoemission matrix element effects in the ARPES spectra, we consider a non-trivial structure factor in ab initio calculations[54,55]. To simulate this effect, we construct a unitary matrix $U(\boldsymbol{k})$:

$$U(\boldsymbol{k}) = \begin{pmatrix} e^{i\boldsymbol{k}\cdot\boldsymbol{r}_1} & \cdots & 0 \\ \vdots & \ddots & \vdots \\ 0 & \cdots & e^{i\boldsymbol{k}\cdot\boldsymbol{r}_n} \end{pmatrix},$$

where $\boldsymbol{k}$ is the momentum vector, $\boldsymbol{r}_i$ is the real space coordinates of the $i$th atom in the original Bi/SnSe(S) unit cell. We simulate the unfolded band structures of Bi/SnSe by applying this unitary matrix to the tight-binding Hamiltonian, $U(\boldsymbol{k})H(\boldsymbol{k})U(\boldsymbol{k})^{\dagger}$. The details of the unfolding procedure are described in the work by W. Ku et al.[56].

The quasiparticle interference pattern was calculated based on Green's function method by using the spin-dependent scattering probability (SSP) method[57,58], which can be written as

$$J_s(q) = \frac{1}{2} \sum_k \sum_{i=0,1,2,3} \rho_i(k)\rho_i(k+q),$$

where $\rho_0(k) = \mathrm{Tr}[G(k)]$ is the total spectral density and $\rho_i(k) = \mathrm{Tr}[\sigma_i G(k)]$ is the spin density, in which $G(k) = [\omega + i\eta - H(k)]^{-1}$ is Green's function of the system and $\sigma_{i=1,2,3}$ are the Pauli matrices for spin.

## Data availability

The authors declare that the data supporting the findings of this study are available within the paper and its Supplementary Information files. The data that support the findings of this study are available from the corresponding authors upon request.

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

## Acknowledgements

The work at the University of Missouri was supported by the U.S. Department of Energy, Office of Science, Office of Basic Energy Sciences, Division of Materials Science and Engineering, under Grant No. DE-SC0024294. G.B. was supported by the Gordon and Betty Moore Foundation, grant DOI:10.37807/gbmf12247. The experimental work was primarily led and supported by the U.S. Department of Energy, Office of Science, the National Quantum Information Science Research Centers, and Quantum Science Center (Q.L., R.G.M., H.J., A.-P.L. were supported). T.-R.C. was supported by the 2030 Cross-Generation Young Scholars Program from the National Science and Technology Council (NSTC) in Taiwan (Program No. MOST111-2628-M-006-003-MY3), National Cheng Kung University (NCKU), Taiwan, and National Center for Theoretical Sciences, Taiwan. This research was supported, in part, by the Higher Education Sprout Project, Ministry of Education to the Headquarters of University Advancement at NCKU. T.-R.C. thanks the National Center for High-performance Computing (NCHC) of National Applied Research Laboratories (NARLabs) in Taiwan for providing computational and storage resources. H.L. acknowledges the support by Academia Sinica in Taiwan under grant number AS-iMATE-113-15. T.-C.C. was supported by the U.S. Department of Energy, Office of Science, Office of Basic Energy Sciences, Division of Materials Science and Engineering, under Grant No. DE-FG02-07ER46383. R.G.M. and M.B. were supported by the U.S. Department of Energy, Office of Science, Basic Energy Sciences, Materials Science and Engineering Division. The STS and QPI works were conducted at the Center for Nanophase Materials Sciences (CNMS), which is a U.S. Department of Energy (DOE), Office of Science User Facility at Oak Ridge National Laboratory.

## Author contributions

G.B., T.-R.C., A.-P.L., and R.G.M. conceived the project. Q.L., G.B., A.R.M., and M.B. synthesized the epitaxial samples. Q.L. did the room-temperature STM measurements with assistance from J.C., C.C., and X.Z. Q.L., M.B., and R.G.M. performed the regular and spin-resolved

ARPES experiments. A.-P.L. and H.J. performed the low-temperature STM, dI/dV, and QPI measurements. P.V.S.R., T.-R.C., H.L., Y.-T.Y., H.-J.T., C.-H.T., P.-Y.Y., S.-W.L., and Q.L. performed the first-principles calculations and simulations. S.A.Y. and W.W. built the effective Hamiltonian. G.B., T.-R.C., S.A.Y., T.-C.C., A.C., and G.V. performed data analysis and theoretical discussions.

## Competing interests

The authors declare no competing interests.
