## [Peer Review File · Nature Communications]

REVIEWER COMMENTS

Reviewer #1 (Remarks to the Author):

I have read the reply of the authors and followed the changes. I thank the authors for their good work. I suggest to publish as is.

Reviewer #2 (Remarks to the Author):

In their resubmission, the authors have rewritten the manuscript considerably in an effort to address the criticism raised by myself and the other referees. I appreciate that effort, and feel that the manuscript has improved quite a bit. Some erroneous or misleading statements were removed, including claims of applicability, which I still find not to be substantiated by the present data.

Now, the manuscript presents careful experiments on a specific material that realizes a 2D Weyl semimetal. The shown data supports the claims, and compares well with the theory.

As the authors acknowledge, their 2D Weyl semimetal is not really a phase, but a fine-tuned situation. Despite that, the authors still talk of their 2D Weyl semimetal as being a "phase". This is wrong: it's a critical situation at the transition between two extended phases.

This point is not emphasised enough in the current manuscript. For example the figure showing the evolution "Gapped 2D Dirac \rightarrow Rashba Splitting \rightarrow Gapless 2D Weyl", is not represented in the main text. Similarly, statements made in the replies, e.g. "It is technically challenging to realize the 2D Weyl semimetal for it is an "accidental" critical phase that is proximal to trivial insulators and quantum spin Hall insulators. To reach this critical phase, one needs to tune a coupling parameter to the critical value." are not fully represented in the main text.

Instead, the main text now states that 2D Weyl semimetals are a new topological state. They underpin this with the notion of a winding number. This indeed is technically not wrong, but the winding number is only "quantized" to $\pm\pi$ if the gap vanishes. There is no topological protection against small perturbations.

Given all of the above, I find that the framing of the results as showing a novel topological phase is simply overselling. It's not a phase, nor is it topologically protected in a way similar to the neighbouring true phases. Instead, the authors perform very nice and exhaustive experiments, and

their data is constant with realising a fine-tuned state at the transition between two gapped topological phases. Given this assessment, I do not support publication of the manuscript in Nature Communications in the present form.

Reviewer #3 (Remarks to the Author):

The authors have carefully considered my concerns and made substantial effort to answer my questions in a proper way. Some additional STM and ARPES experiments were performed and results were included in the manuscript that certainly increased its quality.

I recommend the manuscript for publication.

Referee 2:

In their resubmission, the authors have rewritten the manuscript considerably in an effort to address the criticism raised by myself and the other referees. I appreciate that effort, and feel that the manuscript has improved quite a bit. Some erroneous or misleading statements were removed, including claims of applicability, which I still find not to be substantiated by the present data.

Now, the manuscript presents careful experiments on a specific material that realizes a 2D Weyl semimetal. The shown data supports the claims, and compares well with the theory.

As the authors acknowledge, their 2D Weyl semimetal is not really a phase, but a fine-tuned situation. Despite that, the authors still talk of their 2D Weyl semimetal as being a "phase". This is wrong: it's a critical situation at the transition between two extended phases.

This point is not emphasised enough in the current manuscript. For example the figure showing the evolution "Gapped 2D Dirac \rightarrow Rashba Splitting \rightarrow Gapless 2D Weyl", is not represented in the main text. Similarly, statements made in the replies, e.g. "It is technically challenging to realize the 2D Weyl semimetal for it is an "accidental" critical phase that is proximal to trivial insulators and quantum spin Hall insulators. To reach this critical phase, one needs to tune a coupling parameter to the critical value." are not fully represented in the main text.

Instead, the main text now states that 2D Weyl semimetals are a new topological state. They underpin this with the notion of a winding number. This indeed is technically not wrong, but the winding number is only "quantized" to $\pm\pi$ if the gap vanishes. There is no topological protection against small perturbations.

Given all of the above, I find that the framing of the results as showing a novel topological phase is simply overselling. It's not a phase, nor is it topologically protected in a way similar to the neighbouring true phases. Instead, the authors perform very nice and exhaustive experiments, and their data is constant with realising a fine-tuned state at the transition between two gapped topological phases. Given this assessment, I do not support publication of the manuscript in Nature Communications in the present form.

Authors: We thank the referee for the careful review and valuable comments. We agree with the referee that the Weyl fermion-like quasiparticle is a new “quantum state” rather than a new “phase”. We have toned down in the revision as suggested by the referee.

“This point is not emphasised enough in the current manuscript. For example the figure showing the evolution "Gapped 2D Dirac -> Rashba Splitting -> Gapless 2D Weyl", is not represented in the main text. Similarly, statements made in the replies, e.g. "It is technically challenging to realize the 2D Weyl semimetal for it is an “accidental” critical phase that is proximal to trivial insulators and quantum spin Hall insulators. To reach this critical phase, one needs to tune a coupling parameter to the critical value." are not fully represented in the main text.”

We have moved the discussions on the evolution "Gapped 2D Dirac -> Rashba Splitting -> Gapless 2D Weyl" and the “accidental” critical phase that is proximal to trivial insulators and quantum spin Hall insulators to Supplementary Information. Two statements about those discussions have been added to the text.

On page 5, right after the description of Rashba splitting-induced Weyl states, a statement *“The 2D Weyl states generated through this method represent a fine-tuned critical state at the transition between two topologically distinct gapped phases. A detailed discussion on the formation mechanism of 2D Weyl states can be found in Supplementary Information-Section III”* has been added.

On page 8, A statement *“It is worth noting that, in addition to the Fermi string bands, there are additional edge state bands located within the bulk band gap (in Fig. 4a). The existence of those additional edge state bands can be attributed to the fact that the 2D Weyl semimetal is at a critical point in connection to two topologically distinct insulator phases. Please see Supplementary Information-Section XII for a detailed discussion”* has been added.

“Instead, the main text now states that 2D Weyl semimetals are a new topological state. They underpin this with the notion of a winding number. This indeed is technically not wrong, but the

winding number is only "quantized" to $\pm\pi$ if the gap vanishes. There is no topological protection against small perturbations."

We agree with the referee that the winding number is quantized only in the gapless case. There is no topological protection against small perturbations. On the other hand, the gapless state with quantized winding number shows a topological nature of "bulk-boundary correspondence" as demonstrated in our experiments.

The changes are summarized as follows,

(1) In the abstract, "represents a novel phase of matter" \rightarrow "represents a novel topological matter"

(2) On page 5, right after the description of Rashba splitting-induced Weyl states, a statement "*The 2D Weyl states generated through this method represent a fine-tuned critical state at the transition between two topologically distinct gapped phases. A detailed discussion on the formation mechanism of 2D Weyl states can be found in Supplementary Information-Section III*" has been added.

(3) On page 8, A statement "*It is worth noting that, in addition to the Fermi string bands, there are additional edge state bands located within the bulk band gap (in Fig. 4a). The existence of those additional edge state bands can be attributed to the fact that the 2D Weyl semimetal is at a critical point in connection to two topologically distinct insulator phases. Please see Supplementary Information-Section XII for a detailed discussion*" has been added.

REVIEWERS' COMMENTS

Reviewer #2 (Remarks to the Author):

I thank the authors for the revision of the manuscript. My concerns and criticism were fully accepted, and the manuscript was modified to reflect this criticism. In particular, it is now clear that the Weyl state the authors concentrate on is indeed a critical state.

As written in my last report, the data is very nice, and the result obtained is quite non-trivial and interesting (irrespective of the state being fine-tuned). Given that the data now seems to be appropriately described and contextualised, and that I still believe the results will be interesting to a broad community, I now recommend publication of the paper in the current form.